# Swimbladder Function in the European Eel *Anguilla anguilla*

Bernd Pelster [1,2]

1   Institute for Zoology, Universität Innsbruck, 6020 Innsbruck, Austria; bernd.pelster@uibk.ac.at;
    Tel.: +43-512-5075-1860; Fax: +43-512-5075-1899
2   Center for Molecular Biosciences, Universität Innsbruck, 6020 Innsbruck, Austria

**Abstract:** Eels use the swimbladder for buoyancy control. The ductus pneumaticus connecting the esophagus with the swimbladder is closed soon after initial opening of the swimbladder in the glass eel stage, so that eels are functionally physoclist. Subsequent filling of the swimbladder is achieved by activity of gas gland cells in the swimbladder epithelium and countercurrent concentration in the rete mirabile. Gas gland cells produce and release lactic acid and $CO_2$. In blood, acidification induces a release of oxygen from the hemoglobin (Root effect). The resulting increases in $PO_2$ and $PCO_2$ provide diffusion gradients for the diffusion of oxygen and $CO_2$ into the swimbladder, the main gases secreted into the swimbladder. In addition, the partial pressure of these two gases remains elevated in venous blood leaving the swimbladder epithelium and returning to the rete mirabile. Back-diffusion from venous to arterial capillaries in the rete results in countercurrent concentration, allowing for the generation of high gas partial pressures, required for filling the swimbladder under elevated hydrostatic pressure. The transition of the yellow eel to the silver eel stage (silvering) is accompanied by a significant improvement in swimbladder function, but swimbladder volume cannot be kept constant during the daily vertical migrations silver eels perform during their spawning migration back to the spawning grounds in the Sargasso Sea. Infection of the swimbladder with the nematode *Anguillicola crassus* significantly impairs the function of the swimbladder as a buoyancy organ.

**Keywords:** gas gland; countercurrent exchange; metabolism; Root effect; *Anguilla anguilla*; *Anguillicola crassus*; rete mirabile; hemoglobin; oxygen transport





## 1. Introduction

Eels are usually considered catadromous fish spawning in the marine habitat, and spending most of their life cycle in freshwater systems. It has been shown, however, that some eels may skip the freshwater phase completely and stay in coastal water, or move between brackish water and freshwater (semi-catadromous behavior) [1]. The spawning area of the European eel *Anguilla anguilla* (L. 1758) is the Sargasso Sea. In a recent tracking experiment, it was shown for the first time that adult eels released near the Azores swim to the Sargasso Sea [2]. Tagged eels released near the European coast so far could not be tracked all the way down to the Sargasso Sea [3–6]. Fertilized eggs develop into a larvae named Leptocephalus [7], and later it was realized that the Leptocephali are the larvae of the European eel [8]. The Leptocephali drift with the Gulf stream [9,10] and reach the European or North African continental slope after a journey of about 7 months to 2 years [10]. Before entering the European freshwater system, Leptocephali metamorphose into glass eels. The translucent Leptocephali do not have a swimbladder and appear to be positively buoyant with overall densities of 1.028–1.043 g·mL$^{-1}$ [11]. This low density value appears to be due to a high concentration of glycosaminoglycans in the translucent extracellular matrix. The swimbladder develops and is first inflated in glass eels. The glass eels then develop into so-called yellow eels, which typically spend 5–25 years in the European freshwater system [9,12].

The onset of the spawning migration requires preparation for the transition from freshwater to sea water. This occurs with a process called silvering, originally considered

to be a second metamorphosis, but based on endocrine activity, it appears to be more like a puberty [13]. Silvering includes a number of physiological and morphological changes, including changes in ventral color from yellow to silver, and a significant darkening of the dorsal side. Fat stores and eyes are enlarged [12,14,15], and in the swimbladder, the length of the rete mirabile is increased [16–18]. An increased guanine incrustation has been observed in the American eel [19].

This review summarizes our current knowledge on swimbladder function in the different developmental stages of the European eel. The eel swimbladder has attracted the attention of scientists for over 100 years. Because of the bipolar rete mirabile, which allows taking blood samples in front of the rete and between the rete and the gas gland cells, eels have become a model for swimbladder function. Much information has been gained in a remarkable number of studies, but eels remain a mystery, with a large number of open questions.

## 2. Opening of the Swimbladder in Glass Eel

The swimbladder develops as a dorsal outgrowth of the esophagus. Leptocephali do not have a swimbladder. The first sign of swimbladder development is detected in early metamorphic stages to the glass eel [20]. In many glass eels caught on arrival along the European Coast, the swimbladder does not yet contain any gas, but is filled with surfactant [20]. Surfactant is required to reduce surface tension. Its presence has also been shown in yellow eel swimbladder, and it is produced and secreted by gas gland cells [21]. In early glass eels, the epithelial gas gland cells do not show the extensive basolateral labyrinth, characteristic for gas gland cells of yellow and silver eels (Figure 1). The intimate connection between gas gland cells and blood capillaries also is not yet visible [20]. The histological appearance of the gas gland cells therefore suggested that the initial filling of the swimbladder with gas, observed in the early glass eel stage, is not achieved by secretory activity of these cells, but by gulping air or by taking up small gas bubbles from the water [20]. Within 3 to 4 months of development after the initial inflation, the connective tissue of the swimbladder and the gas gland cells differentiate to the adult state present in yellow eels with an extensive basolateral labyrinth, which is in close contact with swimbladder capillaries [20]. The connection between esophagus and the swimbladder, the ductus pneumaticus, is functionally closed so that the eel cannot gulp air at the water surface. The ductus pneumaticus is used as the resorbing section of the swimbladder, separated from the secretory section via a sphincter muscle, located between the two retia mirabilia [22].

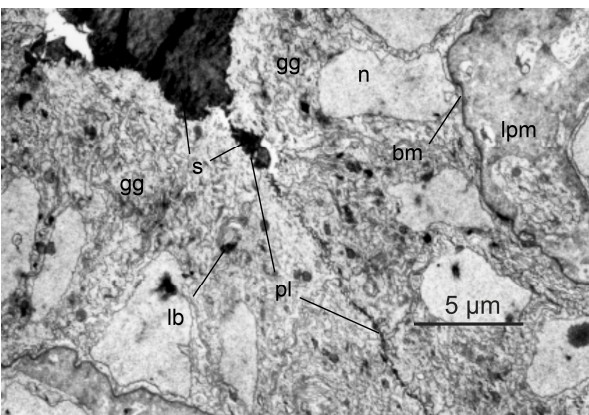

**Figure 1.** Histology of glass eel swimbladder. In Leptocephali developing into glass eels, the swimbladder initially is filled with surfactant; opposing gas gland cells appear to be separated by a thin layer of surfactant. Gas gland cells do not yet express an extensive basolateral labyrinth and the connection between gas gland cells and blood capillaries is not as tight as in yellow eels. gg, gas gland cell; s, surfactant; n, nucleus; pl, plasma membrane; lb, lamellar body; bm, basal membrane; lpm, lamina propria mucosae. Modified after [20].

## 3. Swimbladder Function in Yellow Eels

In the European freshwater system consisting of rivers and lakes, eels experience a limited depth range. Hydrostatic pressure increases by one atmosphere for every ten meters of water depth, so the range of hydrostatic pressures experienced in freshwater is limited. Nevertheless, the swimbladder wall is flexible, and therefore, according to Boyle's law, in a fish descending from the water surface to a water depth of 10 m, it results in a 50% decrease in swimbladder volume. However, neutral buoyancy, a status in which a fish can stay at a certain water depth without any swimming movement, requires a constant swimbladder volume, exactly compensating for the high density of other fish tissues, which exceeds water density [23,24]. The compression of the swimbladder encountered when descending into deeper water therefore must be compensated by gas secretion to keep the swimbladder volume constant. Ascending closer to the water surface, in turn, requires the removal of gas to avoid an increase in swimbladder volume.

Although eels are anatomically physostome, the ductus pneumaticus, the connection between the esophagus and the swimbladder, is functionally closed soon after the original opening of the swimbladder [9,22]. The yellow eel cannot gulp air at the surface and is functionally a physoclist fish. The ductus pneumaticus is used as a resorbing part of the swimbladder. It is a thin-walled, almost translucent gas cavity, highly vascularized, draining into the main venous system. Via a sphincter muscle located between the two retia mirabilia, gas from the secretory section can be transferred to the resorbing part. Along partial pressure gradients, gases then diffuse from this resorbing section to the blood and the venous circulatory system.

The wall of the secretory swimbladder has a silvery appearance due to guanine incrustation in connective tissues, and the epithelium consists of gas gland cells. Arterial blood supply to the secretory swimbladder passes a remarkable countercurrent system, the rete mirabile or red body. In the rete mirabile, the swimbladder artery gives rise to several tens of thousands of capillaries, running in parallel for a distance of several millimeters [25]. The capillaries reunify, forming two or three larger arterial vessels supplying the gas gland cells. Venous return to the rete again forms several tens of thousands of venous capillaries running in parallel and surrounding the arterial capillaries in the rete. The diffusion distance between arterial and venous capillaries is in the range of only one to two micrometer [25]. This arrangement of blood vessels allows taking blood samples in front of the countercurrent system, i.e., at the heart pole, but also between the rete and the gas gland cells, i.e., at the swimbladder pole. Thus, the role of the rete mirabile and the function of gas gland cells can be analyzed separately. This explains why the eel became a model species for the analysis of swimbladder function in physoclist fish [26]. In many other species, this separation of rete and gas gland cell function is not possible because of the close and intimate connection between the two structures [27,28].

In situ studies and experiments with primary cultured gas gland cells confirmed that gas gland cells are specialized for the production of acidic metabolites with a low fraction of aerobic metabolism, although they are chronically exposed to high oxygen partial pressures. A large fraction of glucose taken up from the blood is converted into lactic acid [29–32]. Some glucose is also shifted to the pentose phosphate shunt, where the activity of 6-phosphogluconate dehydrogenase results in the production of $CO_2$, by far exceeding the amount of $CO_2$ produced in the aerobic metabolism [31,33,34] (Figure 2). The activity of the pentose phosphate shunt also results in the generation of NADPH, an important reduction equivalent involved in the degradation of reactive oxygen species.

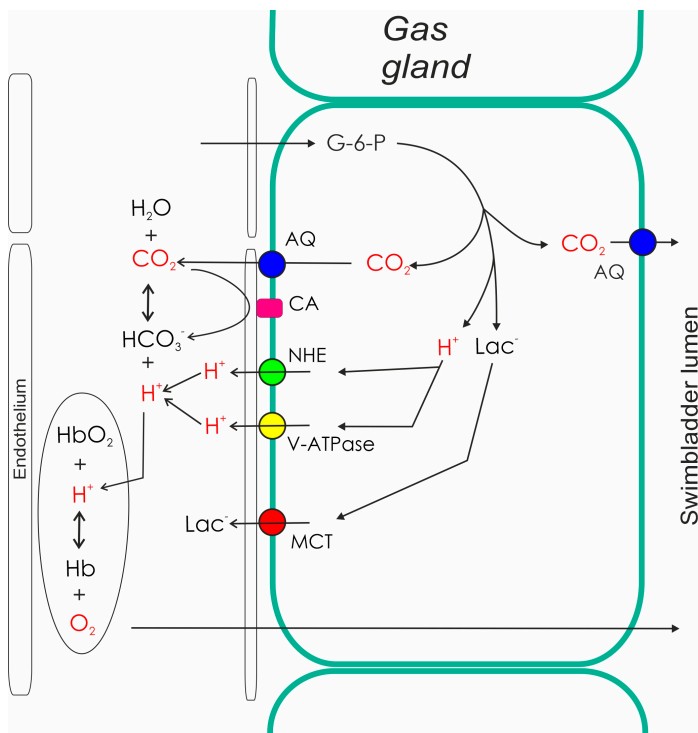

**Figure 2.** Secretory activity of gas gland cells. The metabolism of gas gland cells of the European eel depends on blood glucose, which is converted to lactic acid (lactate anion and a proton), and used for the production of $CO_2$. $CO_2$ production in the pentose phosphate shunt by far exceeds the amount of $CO_2$ produced in the aerobic metabolism. Protons, $CO_2$, and lactate are released into the blood, where the acidification switches on the Root effect. The resulting increase in $PO_2$ provides the partial pressure gradient for the diffusion of oxygen into the swimbladder. The production of $CO_2$ in gas gland cells also generates a partial pressure gradient towards the swimbladder lumen, allowing for the secretion of $CO_2$ into the swimbladder. AQ, aquaporin; CA, carbonic anhydrase; G-6-P, glucose-6-phosphate; Hb, hemoglobin; Lac, lactate; NHE, sodium proton exchanger; MCT, monocarboxylate carrier.

Pharmacological studies indicated that gas gland cells produce and release lactate into the blood stream by a monocarboxylate carrier [35], and analysis of the transcriptome as well as of the proteome of these cells confirmed the presence of monocarboxylate carriers [36,37]. Proton secretion is achieved via sodium proton exchange proteins (NHE-proteins) and a proton ATPase (V-ATPase). V-ATPase subunits and several sodium proton exchange proteins, including NHE1 and NHE2, have been identified in the proteome and in the transcriptome [36–39], and the inhibition of NHE proteins and of V-ATPase both resulted in a significant decrease in acid secretion in isolated gas gland cells [35,40]. Figure 2 summarizes the activity of gas gland cells based on data obtained from the European eel.

Due to the production of $CO_2$ in gas gland cells, the highest $PCO_2$ is expected in these cells, providing a driving force for the diffusion of $CO_2$ into the swimbladder lumen as well as into the blood stream (Figure 2). A membrane-bound carbonic anhydrase has been detected in gas gland cells, establishing a rapid equilibrium between $CO_2$ and bicarbonate in swimbladder blood, and the inhibition of carbonic anhydrase activity reduced the rate of acid secretion [35,41]. In a recent study, the presence of aquaporin 1 was demonstrated in apical and basolateral membranes of gas gland cells, and also in endothelial cells of the swimbladder [42]. In a physoclist swimbladder, water movements between the gas cavity and the surrounding tissue do not make sense. Aquaporin 1 is known to be a water channel, but to be also permeable to $CO_2$ [43–45]. Therefore, it was suggested that these aquaporins act as a $CO_2$ channel [42], facilitating the diffusion of $CO_2$ into the swimbladder as well as into the blood.

Acidification of the blood during the passage of the gas gland cells causes a decrease in the oxygen carrying capacity of the hemoglobin, switching on the Root effect [46–49]. The acid-induced partial deoxygenation of hemoglobin results in a significant increase in $PO_2$ in blood passing the gas gland cells. Thus, a partial pressure gradient is established, driving the diffusion of oxygen into the swimbladder. It also assures that the $PO_2$ in venous blood returning to the rete mirabile exceeds the $PO_2$ of arterial blood leaving the rete.

The activity of the gas gland cells thus induces an initial increase in gas partial pressures in swimbladder capillaries, the so-called single concentrating effect [50]. In a second step, this initial increase in gas partial pressure is multiplied by back-diffusion and countercurrent concentration in the rete mirabile [46,50]. Back-diffusion of oxygen and $CO_2$ from venous to arterial capillaries in the rete mirabile of the European eel resulted in a 7-fold increase in $PO_2$ and an 8-fold increase in $PCO_2$ during arterial passage of the rete mirabile [51].

Gas molecules diffuse through membranes, and the rete mirabile was considered to be a passive exchange system [50]. However, analysis of water and lactate movements in the rete provided the first evidence that not only oxygen and $CO_2$, but also lactate diffuses back to the arterial side in the eel rete [52]. A detailed study of the transcriptome and the proteome of rete capillaries revealed the expression of a large number of membrane transport proteins and several membrane ATPases, including $Ca^{2+}$-ATPase, V-ATPase, and $Na^+/K^+$-ATPase [53]. This suggests that ion transport and proton transport mechanisms indeed contribute to the countercurrent exchange and support the back-diffusion of lactate and protons to the arterial side of the rete in order to switch on the Root effect during the arterial passage of the rete. The data also revealed aquaporin expression in the rete. Because no water shift was detected in the rete, this suggests that aquaporin facilitates the diffusion of $CO_2$ not only in gas gland cells but also in the rete, supporting acidification of the blood in order to switch on the Root effect [53].

## 4. The Effect of Silvering

Silvering, which has been considered to be a puberty-like transition [13], is characterized by remarkable morphological and physiological changes. While the dorsal skin darkens, the ventral skin becomes silvery in preparation for the migration in the open ocean. This counter-shading is known to occur in many pelagic fish [54]. Eyes are significantly enlarged, and the diameter of the eye is among other parameters used to describe the silvering status of eels [55]. Neuromasts develop near the lateral line and fat stores increase by up to 28%. Eels do not feed during their spawning migration, and the alimentary tract degenerates [3,9,12,14,15].

An elongation of the rete mirabile has been described for the American eel [16], and a reduction in the gas permeability of the swimbladder wall occurs by increasing wall thickness and guanine deposition [18]. Cholesterol in cell membranes also contributes to a low gas permeability, but compared to yellow eels, in silver eels the cholesterol content was not increased [42]. Comparison of the transcriptome of yellow and silver European eels indicated significant expression changes for genes related to the extracellular matrix, supporting the conclusion that gas permeability of the swimbladder wall is reduced in silver eels [37]. Moreover, genes related to the degradation of reactive oxygen species (ROS) were over-expressed in silver eels. Due to greater depth encountered in the open ocean, higher oxygen pressures must occur in the swimbladder, probably coinciding with an increased level of ROS production [37]. Genes related to glucose transport were elevated in their transcription level in silver eels, while the expression of glycolytic enzymes was not enhanced [37].

An elevated expression of glucose transport and of monocarboxylate transport proteins was confirmed by analysis of the proteome [36]. The proteome also revealed expression of glucose-6-phosphate dehydrogenase, a crucial enzyme of the pentose phosphate shunt. A large number of enzymes related to ROS defense, including catalase, superoxide dismutase, and peroxiredoxin, were detected in the proteome, and glutathione peroxidase (GPX-7) was

upregulated [36]. Measurement of enzyme activities confirmed an elevated ROS defense capacity in silver eels [56]. A summary of the basic functional characteristics of a yellow eel swimbladder and of the improvements observed during silvering is presented in Figure 3.

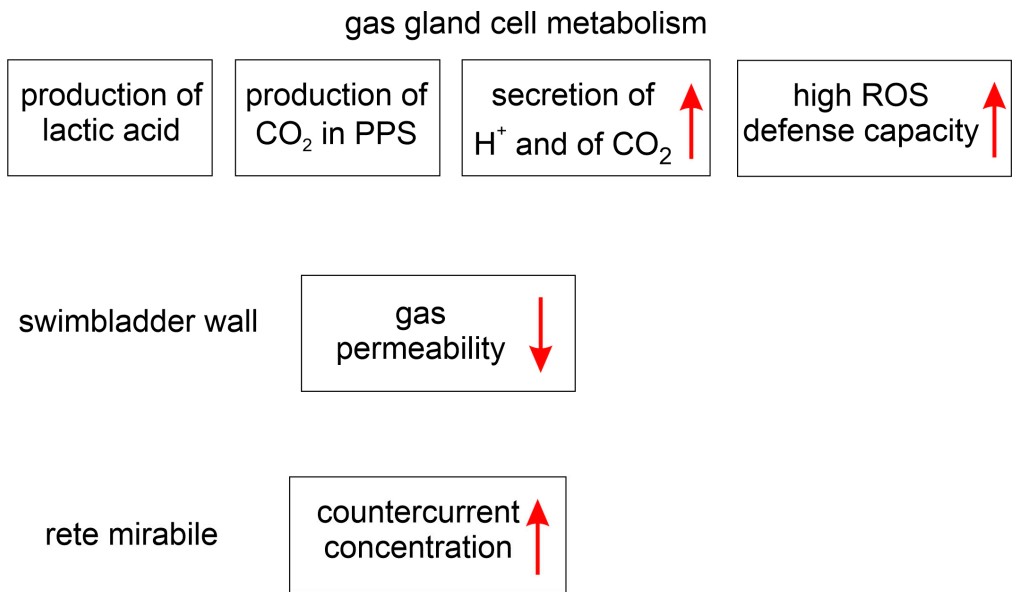

**Figure 3.** Functional characteristics of eel swimbladder and improvements related to silvering, indicated by a red arrow. Silvering results in an improvement of the secretory activity of gas gland cells, and in an improvement of the ROS defense capacity. Gas permeability of the swimbladder wall is reduced; the countercurrent concentrating capacity of the rete mirabile is enhanced. PPS, pentose phosphate shunt, ROS, reactive oxygen species.

### 5. Swimbladder Function during the Spawning Migration

Many studies have attempted to track tagged eels during their spawning migration to the Sargasso Sea. Depending on the starting point in Europe, the spawning migration covers a distance of 5000 to 6000 km, but eels could successfully be tracked for up 1000 km, and only occasionally for up to 2000 km [4–6]. Only eels released at the Azores, i.e., about halfway between Western Europe and the Sargasso Sea, could be tracked to the Sargasso Sea [2]. An unexpected result of all tracking studies was the observation that eels perform diurnal vertical migrations. While in the Baltic or in the North Sea, the depth range is below 100 m, beyond the continental shelf, eels swim at depths of about 200–300 m at nighttime, and at depths of 600–800 m during daytime. Eels have even been recorded at depths below 1000 m [5,6,57,58]. Eels do not feed during their spawning journey [9,59]; therefore, these diurnal excursions cannot be related to feeding. Thermoregulation and predator avoidance have been discussed as possible reasons [3,58,60], and a recent study suggested that eels follow an isolume with these vertical movements [4]. It is generally expected that the journey takes about 4 to 6 months, but recent considerations indicate that it may also take more than a year [4,59,61]. Experiments indeed revealed that eels are able to swim for several months and can cover a distance of up to 5500 km in a swim tunnel without feeding [62,63]. Body composition of eels after six months in a swim tunnel was not different from control eels, indicating that eels used fat, protein, and carbohydrates in the same proportion [63]. Based on the decrease in energy reserves and on oxygen consumption, optimal swimming speed and the cost of transport have been assessed, and all studies consistently show that eels swim very efficiently, with the cost of transport being significantly lower than in salmonids [59,61,63,64].

These studies have been performed under atmospheric pressure. The measuring of oxygen uptake of eels kept under elevated hydrostatic pressure for prolonged periods revealed decreased oxygen consumption as compared to values recorded under atmospheric pressure, suggesting that the actual costs of transport during the spawning migration

may even be lower than calculated based on swim tunnel experiments under atmospheric pressure [65]. These data clearly show that eels are able to successfully complete the spawning migration, and preserve sufficient energy for gamete production and successful reproduction.

With the diurnal vertical migration, eels experience significant changes in hydrostatic pressure. While at a depth of 300 m, hydrostatic pressure amounts to 31 atm, at 800 m it increases to 81 atm, resulting in proportional compression of the flexible walled swimbladder according to Boyle's law. With a reduced swimbladder volume, eels would become negatively buoyant, if the eels would not be able to fully compensate for the increase in hydrostatic pressure by secreting gas in order to keep the swimbladder volume constant. The depth records obtained from tagged eels reveal that the descent from the depths at night to greater depths in the daytime takes no more than 1 to 2 h, and the same time is required for the return to the depth range at the beginning of the night [4,6]. Accordingly, at constant temperature, about 2.500 mL of gas would be required for a 1 kg eel to keep the swimbladder volume constant during a descent from 300 m to 800 m [66]. In the European yellow eel, gas secretion rates did not exceed 1–2 mL·h$^{-1}$ [51,67]. Values below 1 mL·h$^{-1}$ have also been reported for the American yellow eel *Anguilla rostrata*, and in silver eel, the rate of gas secretion increased to up to 3 mL·h$^{-1}$ [18]. These values are far too low to keep the swimbladder volume constant during the descent from 300 m to 800 m in less than 2 h. Taking into account the oxygen transport capacity of European eel blood and the possible glucose turnover in gas gland cells, it seems unlikely that the required amount of gas can be supplied to keep the volume constant [66].

If the swimbladder would provide neutral buoyancy at a depth of 800 m, the same amount of gas would have to be resorbed and transported in the blood within 1 to 2 h during ascent to avoid increasing positive buoyancy. This also appears unlikely based on gas transport capacities determined in eel blood. Therefore, it can be assumed that the swimbladder may provide neutral buoyancy at the depths encountered at night time, and is compressed during descent in the daytime. In this case, the lack of buoyancy during descent must be compensated by hydrodynamic lift [66].

The changes in swimbladder and rete mirabile structure and in swimbladder metabolism outlined above, however, clearly indicate an improvement in swimbladder function. Additionally, although the rate of gas secretion is not sufficient to keep the swimbladder volume constant during these vertical migrations, there must be some secretory activity. Gas partial pressure in the water hardly increases with depth, so that the partial pressure gradient between swimbladder lumen and the surrounding water increases with depth [68]. Although the gas permeability of the swimbladder wall is low, it certainly is not impermeable to gases, and the diffusional loss of gas through the wall increases with depth along the increasing partial pressure gradient between the lumen and the surrounding tissues. This loss must be compensated for by gas secretion. Moreover, even if a full compensation is not possible, this does not mean that there is no secretory activity at all to achieve at least a partial compensation. Silver eels traveling in the open ocean experience a much larger depth range than yellow eels in freshwater, and this may explain the improved secretory activity detected in silver eels.

## 6. The Nematode *Anguillicola crassus* Impairs Swimbladder Function

In the early 1980s, the swimbladder nematode *Anguillicola crassus* was accidentally introduced to Europe, and within about a decade, a large fraction of the European eel population was infected with this nematode [69–75]. The eel is infected by feeding on intermediate hosts of the nematode (copepods) or diverse paratenic hosts, such as smaller fish species, amphibians, insect larvae, crustacea, or mollusks [73,76]. Eels are infected with L3-stage nematode larvae, which enter the swimbladder wall. Experiments with glass eels revealed that, even at this early stage, eels can be infected when fed with the L3-stage-nematode-infected copepods [77]. In the swimbladder, the histophagic L3 larvae develop into L4-stage larvae and the preadult stage, which enters the swimbladder lumen. Adult nematodes live in the swimbladder lumen and feed on blood from swimbladder

capillaries [78,79]. Reductions in hematocrit and plasma protein content have been observed as a result of this hematophagy [79]. Eggs of the nematode hatch in the swimbladder and leave the swimbladder via the ductus pneumaticus and the gut [73,77,80].

A comparison of the oxygen content of infected and non-infected yellow eels revealed a significant reduction in oxygen content, suggesting a significant impairment of swimbladder function [41] (Figure 4). A heavily infected swimbladder of yellow eels may be almost completely filled with nematodes and dark fluid, containing almost no gas at all. In this situation, the buoyancy function of the swimbladder is completely lost, and hydrodynamic lift will be required to prevent sinking [23,24]. An infection of the swimbladder not only changes the gas composition and the amount of gas in the bladder, but it also results in significant changes in the swimbladder wall. Swimbladder epithelial cells proliferate and the single-layered epithelium thickens, forming a multilayered epithelium. The extent of the basolateral labyrinth is reduced, and the intimate contact to blood capillaries and the polarity of the cells is partially lost [74,80,81]. This results in a significant increase in diffusion distances between blood capillaries and the swimbladder lumen, and also between gas gland cells and the blood, explaining the decrease in oxygen content observed in infected yellow eel swimbladders [41]. In addition, the elasticity of the swimbladder wall is significantly reduced [82].

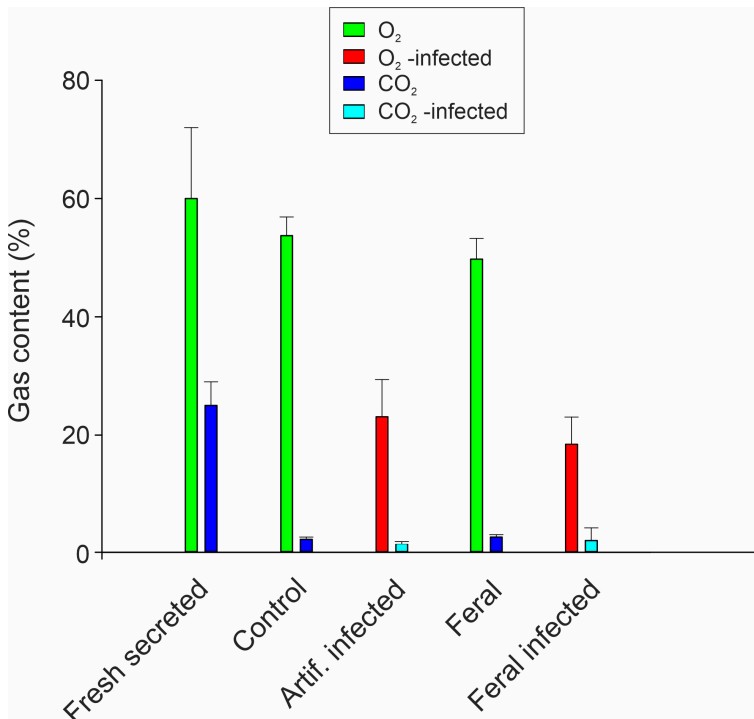

**Figure 4.** Oxygen and $CO_2$ content of healthy and infected eel swimbladder. Oxygen makes up the largest fraction of freshly secreted gas in the eel swimbladder, under steady-state conditions and in the swimbladder of feral eels collected in the wild. The fraction of $CO_2$ is also high in freshly secreted gas, but under steady-state conditions, it is much lower because of the preferential resorption due to the high physical solubility of $CO_2$. In eels artificially infected with the nematode *Anguillicola crassus* and in feral eels with infected swimbladders, the oxygen content is largely reduced, and the $CO_2$ content is also reduced. Data from [67,77].

Infection of the swimbladder in yellow eels resulted in a modification of the mRNA level of more than 1600 genes [38]. Functional annotation based on GO terms revealed that a large number of genes related to metabolism were affected in infected eels. Several transcripts coding for proteins involved in glucose and monocarboxylate transport were elevated in their expression level. The highest number of affected genes, however, were related to the term 'immune response', indicating that yellow eels tried to defend the

nematode. Inflammatory components, complement proteins, and also immunoglobulins were significantly elevated in their mRNA expression level [38]. An increased activity of the immune system has also been detected by testing the antibody response following an artificial infection in both European and Japanese eel [73]. In addition, an elevated non-specific immune response in response to the nematode infection has been observed in wild and farmed European eels [83]. The presence of macrophages in infected swimbladders underlines the initiation of a strong immune response by the nematode [72,84,85].

While a large number of genes were differentially expressed at the mRNA level in infected yellow eel swimbladders, parasite infestation caused only minor expression changes in silver eels, with no apparent changes in the mRNA content of genes connected to glycolytic acid production or ROS defense, both crucial for swimbladder function [38]. Silvering includes remarkable physiological modifications, including the preparation for the transition from freshwater to sea water with appropriate changes in ion and osmoregulation. It also marks the onset of sexual maturation [86,87], although sexual maturity is only achieved at some point during the spawning migration. These changes require a lot of energy, and this may explain the significantly diminished defense response of silver eel swimbladder tissue to the nematode infection.

Comparison of transcriptional changes in non-infected vs. infected swimbladders in silver eels exercising in a swim tunnel under elevated hydrostatic pressure (8 atm) [88] showed a 3-fold greater number of differentially expressed genes in infected eels. In healthy eels, genes with an elevated transcription level were related to glycolytic activity, glucose transport, and carbonic anhydrase expression, suggesting an improvement of the gas secretory activity. Elevated levels of angiopoietin transcripts suggested an improvement of the gas gland cell blood connection, facilitating acid release and switching on the Root effect in swimbladder capillaries [88]. In infected eels, elevated transcriptional levels of glycolytic enzymes including hexokinase 2 point to the activation of glycolysis due to tissue thickening and elevated diffusion distances. Transcriptional changes in infected swimbladders also showed an elevated transcription of immune response genes [88], confirming the results obtained with unexercised yellow and silver eels [37].

Focusing on the physiology and the performance of the swimbladder, a comparison of oxygen consumption of glass eels infected and non-infected with nematodes revealed no significant difference in oxygen consumption [77]. A comparison of the swimming capacity of female silver eels with infected and non-infected swimbladders revealed a reduced swimming capacity in infected eels [89]. Eels with nematode-infected swimbladder had lower cruising speeds, and 27 out of 74 eels swam unsteadily and stopped swimming at comparatively low aerobic swimming speeds. These 27 eels had a significantly higher oxygen consumption in the swim tunnel, and in eels with infected swimbladder, the costs of transport were elevated. In endurance trials, eels with heavily infected swimbladders failed to swim 1000 km [89]. In eels up to 45 cm body length with more than 10 nematodes in the swimbladder, a significant reduction in swimming speed has been observed [90], and silver eels with several adult nematodes in their swimbladder have been shown to avoid accelerating water velocity [91]. This could result in a delayed passage of downstream rapids and thus in a delayed start in the spawning migration.

The study of Simon et al. [92] suggested that nematode infestation does not influence diurnal diving behavior. However, given that the study was conducted only with one single individual infected with three nematodes and a limited depth (60 m), this observation does not exclude the possibility of a negative impact with more parasites or on longer and deeper dives. On the basis of tracking studies, Wysujack et al. [5] speculated that nematode infestation did not preclude daily vertical migrations, although the study did not include the actual infection rate of the silver eels.

Taken together, these data clearly show the tremendous negative impact of the nematode *Anguillicola crassus* on the swimming capacity of eels and on swimbladder function. The nematode infection, therefore, is expected to be a serious threat to successful spawning

migration to the Sargasso Sea, and it appears to be one of the parameters contributing to the current decline in the eel population.

## 7. Future Directions

Although the study of the swimbladder has attracted the attention of scientists for many decades, it remains a mystery. We have learned much about the basic concept of gas secretion, including the secretory activity of gas gland cells, the Root effect, and countercurrent multiplication of the single concentrating effect. However, we know little about the exact transport characteristics of the countercurrent system, the rete mirabile, which, as recently detected, is not only a passive exchange system—transport proteins and especially ATPases may be controlled in its activity. Therefore, it will be interesting to assess possible regulatory mechanisms for the back-diffusion of ions and metabolites in the rete mirabile. Although repeatedly assessed and important, because gas gland cell metabolism and the rate of acid secretion determine the rate of gas deposition [67], the mechanisms controlling gas gland cell activity remain undefined. Still unclear are the role and the performance of the swimbladder during diurnal vertical migrations. It appears unlikely that the swimbladder can provide neutral buoyancy during the rapid descent and ascent associated with these migrations, but the improvements connected with silvering of the eel clearly demonstrate that the swimbladder is of importance during these migrations.

**Funding:** This research was funded by Austrian Fonds zur Förderung der wissenschaftlichen Forschung (FWF), P26363; I2984.

**Conflicts of Interest:** The author declares no conflict of interest.

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
