# Peer review of "Swimbladder Function in the European Eel Anguilla anguilla"

_fishes, doi:10.3390/fishes8030125_

Round 1

Reviewer 1 Report

The paper “Swimbladder function in the European eel Anguilla anguilla” from B. Pelster is well enclosed in the Special Issue “Biology and Ecology of Eels” and summarizes current knowledge on the swimbladder function in the different developmental stages of the European eel.

The review design is well organized and both functional and molecular aspects are well discussed. Although the MS is well written, some sentences result unstructured and difficult to read (see for example: pag. 11, “In a recent study silver eels……that had been exercised”; “Based on one eel……diurnal diving behaviour”). In my opinion, the MS can be accepted for publication after a careful check of typos and style errors.

Moreover, I also suggest to better clarify the advantages of using eel swimbladder as a model for swimbladder function; this will increase the scientific interest of the topic.    

I also have few minor comments:

-       Figure 1: nucleus is indicated in the legend but not in the figure

-       Page 3: Please remove “i.e. the connection between the esophagus and the swimbladder” because already specified.

-       Page 4, last sentence: please check and rephrase

Author Response

Response to reviewer 1

The manuscript has been checked for typos and style errors. Unstructured sentences have been reworded, typos have been eliminated. The reason, why the eel swimbladder is a model for swimbladder function, has been explained more clearly in the introduction.

In Figure 1 the nucleus has been indicated.

i.e. has been removed from page 3, the last sentence of page 4 has been reworded.

Reviewer 2 Report

The paper reviews a large number of publications addressing the swimbladder function of European eel in different developmental stages, during the spawning migration, or in case of parasitic infestation with nematodes.  The paper is well-written, and the information flows in a coherent manner.

However, an improvement of graphical content (in special Fig.3) could upgrade the manuscript.  Some double spaces between words should be corrected.

Page 7 ”cannot be related feeding” change to ”cannot be related to feeding”

Author Response

Figure 3 has been modified and renewed.

The sentence on page 7 has been reworded.